# Antagonism between parasites within snail hosts impacts the transmission of human schistosomiasis

Martina R Laidemitt[1,2]*, Larissa C Anderson[1,2], Helen J Wearing[1,2,3], Martin W Mutuku[4], Gerald M Mkoji[4], Eric S Loker[1,2]

[1]Department of Biology, University of New Mexico, Albuquerque, United States; [2]Center for Evolutionary and Theoretical Immunology (CETI), University of New Mexico, Albuquerque, United States; [3]Department of Mathematics and Statistics, University of New Mexico, Albuquerque, United States; [4]Centre for Biotechnology Research and Development, Kenya Medical Research Institute, Nairobi, Kenya

**Abstract** Human disease agents exist within complex environments that have underappreciated effects on transmission, especially for parasites with multi-host life cycles. We examined the impact of multiple host and parasite species on transmission of the human parasite *Schistosoma mansoni* in Kenya. We show *S. mansoni* is impacted by cattle and wild vertebrates because of their role in supporting trematode parasites, the larvae of which have antagonistic interactions with *S. mansoni* in their shared *Biomphalaria* vector snails. We discovered the abundant cattle trematode, *Calicophoron sukari,* fails to develop in *Biomphalaria pfeifferi* unless *S. mansoni* larvae are present in the same snail. Further development of *S. mansoni* is subsequently prevented by *C. sukari*'s presence. Modeling indicated that removal of *C. sukari* would increase *S. mansoni*-infected snails by two-fold. Predictable exploitation of aquatic habitats by humans and their cattle enable *C. sukari* to exploit *S. mansoni*, thereby limiting transmission of this human pathogen.

*For correspondence:
mlaidemitt@unm.edu

Competing interests: The authors declare that no competing interests exist.

## Introduction

Infectious diseases exist in complex ecological settings and for some disease agents, successful transmission may require sequential colonization of multiple obligatory hosts. This is particularly important for digenetic trematodes because obligatory hosts must overlap in space and time and in sufficient numbers for transmission to be successful (*Dobson, 2004*; *Begon, 2008*; *LoGiudice et al., 2008*). In addition, the environment may harbor a diversity of host species, some of which might serve as alternative competent hosts thus potentially favoring parasite persistence, or conversely, such species may act as non-productive sinks for the parasite thereby diminishing its transmission. As shown by an increasing number of studies, the net effect of biotically complex environments on transmission success for a particular parasite may be hard to predict, particularly for parasites that use multiple hosts to complete their life cycles (*Lafferty et al., 1994*; *Suzán et al., 2009*; *Lafferty, 2012*; *Johnson et al., 2013*; *Salkeld et al., 2013*; *Rohr et al., 2015*; *Frainer et al., 2018*). Furthermore, the impact may be highly contextual, depending on the density of host species, leading to dilution or amplification of transmission (*Luis et al., 2018*).

Another factor influencing the success of a particular parasite is the diversity of other infectious agents present that colonize the same host species, thereby setting up the potential for within-host interactions. Such interactions can range from mutual dependence or facilitation to overt competition and even predation of one parasite by another (*Lim and Heyneman, 1972*; *Combes, 1982*; *Lafferty et al., 1994*; *Hechinger et al., 2011*). The purpose of this study is to combine field observations, experimental infections and use of a mathematical model parameterized by our empirical data

to understand how transmission of the widespread human-infecting trematode *Schistosoma mansoni* is influenced by other trematode species present in the aquatic habitats in and around Lake Victoria. We are particularly interested in those trematode species that share with *S. mansoni* a dependence on *Biomphalaria* snails for their larval development. These additional *Biomphalaria*-dependent trematodes also infect a variety of wild or domestic host species to complete their respective life cycles. Therefore, strong interactions occurring among trematode species in the snail host could have cascading effects on the level of parasitism occurring in many host species, including humans and domestic animals.

*S. mansoni* is the most geographically widespread causative agent of intestinal schistosomiasis, and the Lake Victoria Basin is one of the largest hyperendemic areas of schistosomiasis in the world (*Gouvras et al., 2017*; *Wiegand et al., 2017*). Despite repeated treatments with the anthelmintic praziquantel, children living in villages near the lakeshore often exhibit prevalence of *S. mansoni* infection of >50%, and up to 90% in some areas (*Woodhall et al., 2013*).

A textbook-like portrayal of the *S. mansoni* life cycle highlighting the role of humans and a generic '*Biomphalaria*' in transmission belies a more complex reality in the Lake Victoria Basin in that multiple options exist for transmission. With respect to mammalian definitive hosts, humans are certainly the most important in maintaining transmission (*Colley et al., 2014*), but baboons and common rodents like *Mastomys natalensis* can also play a role and may assume greater significance in ongoing chemotherapy-based control programs targeting human transmission (*Hanelt et al., 2010*; *Catalano et al., 2018*). Likewise, *S. mansoni* in and around the lake also infects three *Biomphalaria* taxa that exist in different habitats. *Schistosoma mansoni* eggs are passed into aquatic habitats in human feces. A miracidium hatches from an egg and if successful in finding and penetrating a *Biomphalaria* snail, then initiates a long-term, proliferative period of asexual development. This development culminates in the production of human-infective cercariae which are released or 'shed' into the surrounding water. These obligatory vector hosts include *B. pfeifferi* in streams and small impoundments, *B. sudanica* along the lakeshore, and *B. choanomphala*, now generally considered to be a distinct ecophenotype of *B. sudanica*, found especially but not exclusively in deeper water (*Magendantz, 1972*; *Standley et al., 2011*; *Zhang et al., 2018*). All three species have been found naturally infected with *S. mansoni*, (*Magendantz, 1972*; *Standley et al., 2011*; *Mutuku et al., 2017*; *Mutuku et al., 2019*).

In addition to exploiting multiple definitive and intermediate host species, *S. mansoni* is, like many infectious agents, influenced in a myriad of ways by the diverse species with which it co-occurs. For instance, several other non-host species of freshwater snails co-occur with *Biomphalaria* and may act as sinks for *S. mansoni* as its miracidia will not develop in these snail species. Likewise, there are other species of digenetic trematode species in Kenya with an obligatory dependence on *Biomphalaria* snails for their larval development and their larval stages may strongly interact with *S. mansoni* larvae for access to the resources offered by these snail hosts. It is well-known from other studies that complex interactions usually dominated by antagonism via predation (and by indirect antagonism) can occur amongst the larvae of trematode species sharing a particular individual snail, and these interactions often have predictable outcomes that can influence patterns of abundance among trematode species in particular communities (*Fernandez and Esch, 1991*; *Lafferty et al., 1994*; *Soldánová et al., 2012*; *Mordecai et al., 2016*).

Relevant to considerations of intra-snail interactions is that not all larval trematodes have the same pattern of development within their host snail. Some like *S. mansoni* develop sac-like sporocysts, whereas others produce rediae possessing a mouth surrounded by a powerful sucker, and a gut (*Schell, 1985*). Rediae can move within the snail and may ingest host tissue, including gonadal tissue (*Lim and Lie, 1969*). For example, the rediae of echinostomes (Echinostomatoidea) in particular tend to be dominant in intramolluscan interactions, with some echinostome species able to produce small, motile rediae specialized for attacking and killing larvae of other trematodes species, including *S. mansoni* sporocysts (*Lim and Heyneman, 1972*; *Moravec et al., 1974*; *Hechinger et al., 2011*). Amphistome trematodes (Paramphistomoidea) also produce rediae, and these tend to have more complex antagonistic or facilitation interactions with other trematodes (*Southgate et al., 1989*; *Spatz et al., 2012*). Sporocyst-producing species including strigeids and schistosomes have also been shown to have more subtle indirect antagonistic effects on larvae of other trematodes mediated by as yet uncharacterized soluble factors (*Basch et al., 1969*). Given the complex nature of the guilds of trematodes, we found to be associated with *Biomphalaria* snails in

Kenya, we hypothesized that some of these non-schistosome trematode species could negatively impact the development and transmission of *S. mansoni* in *Biomphalaria*.

To better understand these interactions, we initiated a survey at two stream sites and one lake site in western Kenya to define the trematode diversity from *Biomphalaria*. Morphological features and molecular markers were used to aid explicit definition of trematode species diversity. We also used sequential observations and experimental infections to reveal the dominance hierarchy among the trematodes that use *B. pfeifferi,* the most widely distributed *S. mansoni* vector in Africa. The same approaches also enabled us to observe striking antagonistic interactions perpetrated on *S. mansoni* by the most common trematode recovered from our habitats, the cattle-transmitted amphistome *Calicophoron sukari*. We then integrated our field survey results and experimental studies into a mathematical model to quantify the influence of *C. sukari* on *S. mansoni* transmission. These studies collectively highlight the importance of animal husbandry practices and of a diverse community of wild animals and the trematodes they support, particularly those dependent on *Biomphalaria*, on downstream transmission of *S. mansoni* infection to people in west Kenya.

## Results

### Field surveys

Upon screening *Biomphalaria* from both stream and lake habitats, several trematode species were found, verified as distinct taxa using morphological features and molecular markers (*Figure 1a*). A total of 19 species were found among 19,914 *B. pfeifferi* (overall prevalence 12.3%) from the perennial Asao Stream, 7 species from 1,136 *B. pfeifferi* (15.5%) from the highly seasonal Kasabong stream, and from Lake Victoria, 21 species of cercariae from 3,369 *B. sudanica* (10.3%), (*Figure 1b*).

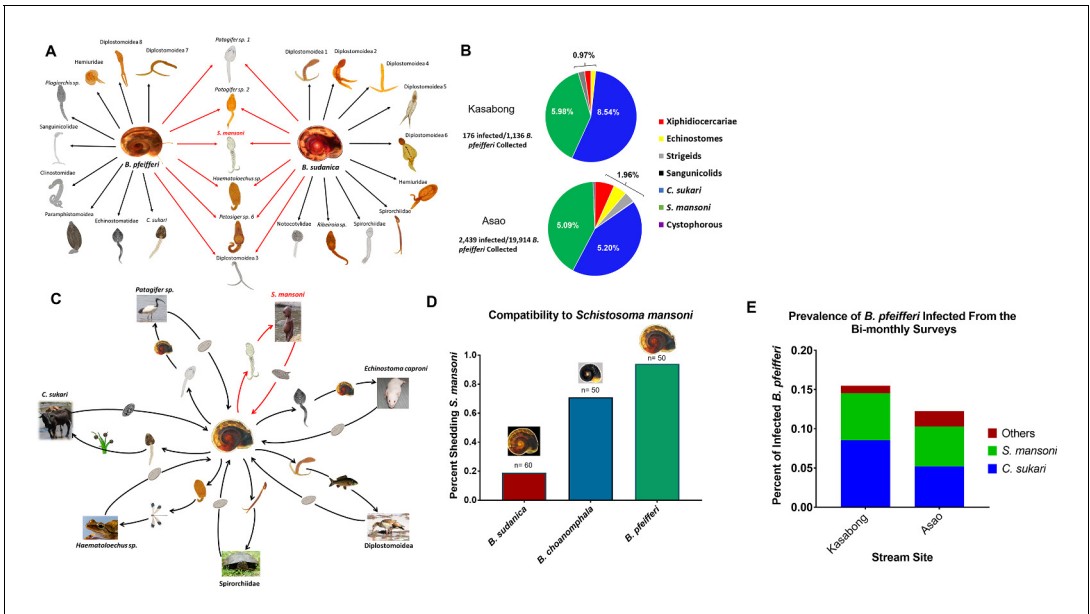

**Figure 1.** Trematode biodiversity in the *S. mansoni* hyperendemic area of western Kenya. (**A**) The different types of trematode cercariae recovered either uniquely from *B. pfeifferi* from streams or from *B. sudanica* from the lakeshore, or that were recovered from both snail species (in red), confirmed with mitochondrial barcodes. (**B**) Kasabong (ephemeral stream) and Asao (perennial stream), the overall prevalence of larval trematode infections in *B. pfeifferi* is shown along with pie charts showing the composition of the trematode infections. Note the large proportions of infections comprised of *C. sukari* or *S. mansoni*. (**C**) A sample of six trematode with inferred life cycles (in some cases directly documented) to point out the variety of invertebrate and vertebrate hosts (and plants) involved in such cycles. (**D**) Peak prevalence of *S. mansoni* (indicated on vertical axis as the percent shedding cercariae) from three different *Biomphalaria* taxa to experimental infection with *S. mansoni* miracidia (five miracida/snail) derived from local school children. (**E**) The prevalence of *C. sukari*, *S. mansoni*, and other trematodes from bimonthly surveys of Kasabong and Asao. Note *C. sukari* is the most abundant trematode in both locations. The turtle, carp and frog images in panel C were reproduced from Pixabay (https://pixabay.com/photos/turtle-animal-wildlife-wild-nature-1517920/, https://pixabay.com/vectors/animal-carp-fish-freshwater-lake-2029698/, and https://pixabay.com/photos/tree-frog-anuran-frog-amphibians-299886/ respectively), under the terms of the Pixabay license (https://pixabay.com/service/license/).

Some species, like *S. mansoni*, were recovered from all habitats, but many were habitat specific, such that in aggregate, 29 taxa were recovered. The life cycles of most of these trematodes have yet to be clarified, but by inferring probable life cycles from relatives with known life cycles (*Yamaguti, 1958*; *Schell, 1985*), a broad variety of invertebrate and wild and domestic vertebrate species are involved as hosts in supporting these life cycles (*Figure 1c*). A portion of the field-collected *Biomphalaria* were also exposed to *S. mansoni* to assess compatibility and we found that all three taxa of *Biomphalaria* were susceptible to *S. mansoni,* but found that *B. pfeifferi* was the most compatible (*Figure 1d*). At Asao and Kasabong, the most common trematode recovered from *B. pfeifferi* was the cattle-transmitted amphistome *Calicophoron sukari* (*Laidemitt et al., 2017*) (1,035/2439 total infections, or 42.4% of all infections at Asao, and at Kasabong 97/176, or 55.1% of all infections) followed closely by *S. mansoni* (1014 of 2439 total infections at Asao or 41.6% of all infections and at Kasabong, 68/176 or 38.6% of all infections) (*Figure 1e*). Both species were also commonly recovered from *B. pfeifferi*, but *C. sukari* was not recovered from *B. sudanica* or *B. choanomphala*. Also noteworthy is the overall prevalence of *S. mansoni* infections at stream sites (5.09% and 5.89%) was much higher than from the lakeshore (0.22%).

## Dominance hierarchy

By observing natural takeover events (*Biomphalaria* shedding one type of cercariae and then later shedding a different type of cercariae only) and through experimental infections, we discovered that at Asao Stream *S. mansoni* occupies an intermediate position in the hierarchy, and echinostomes and *C. sukari* were the more dominant species (*Figure 2*). In general, amphistomes of domestic ruminants like *C. sukari* were more commonly recovered from *B. pfeifferi* from stream sites whereas echinostomes, mostly transmitted by birds (*Laidemitt et al., 2019*), were more likely to be recovered from *B. sudanica* from the lakeshore.

We discovered that if *B. pfeifferi* from stream habitats that were not shedding cercariae of any kind at the time of collection were then exposed to *S. mansoni* miracidia, surprisingly they were almost as likely to subsequently shed *C. sukari* as *S. mansoni* cercariae (p<0.001) (*Figure 3a*). Control snails, otherwise similar except not exposed to *S. mansoni*, were subsequently much less likely to shed cercariae of either species.

## *C. sukari* and *S. mansoni* Experimental Exposures

Three additional experiments were undertaken with laboratory-bred F1 *B. pfeifferi* exposed to *S. mansoni* (originating from primary school children), to *C. sukari* (from cow dung samples collected from the banks of the streams), or to both species in various combinations (*Figure 3b*) and see methods. In control experiments, peak *S. mansoni* shedding was 7 weeks post exposure (86.7%, 92/106 surviving *B. pfeifferi*) and peak *C. sukari* shedding was at 8 weeks post exposure (0.68%, 1/114 surviving *B. pfeifferi*). In control experiments, *S. mansoni* shedding prevalence was 75.9% (63/83 surviving snails) 10 weeks post exposure and *C. sukari* shedding prevalence was 1.3% (1/79 of surviving snails). In the experiment 'Sm first then two weeks later Cs,' 24.2% (16/66) surviving *B. pfeifferi* shed *S. mansoni* and 21.2% (14/66) shed *C. sukari* cercariae. In the experiment 'Cs then two weeks later Sm,' 35.1% (26/74) of surviving *B. pfeifferi* shed *C. sukari* and 2.7% (2/74) shed *S. mansoni* cercariae. In the simultaneous experiment 9.2% (7/76) surviving *B. pfeifferi* shed *S. mansoni* and 15.8% (12/76) shed *C. sukari* cercariae. The results demonstrated that *S. mansoni* miracidia were capable of infecting *B. pfeifferi* on their own as expected (*Mutuku et al., 2014*), but *C. sukari* miracidia were poorly infective to *B. pfeifferi* on their own. From these co-exposure experiments the prevalence of *B. pfeifferi* shedding *S. mansoni* was significantly less than in the simultaneous (p=0.0478) and 'Cs then two weeks later Sm' experiments (p=0.0292) compared to *S. mansoni* controls. In comparison, *C. sukari* prevalence was significantly higher (p=0.0192) in the 'Cs then two weeks later Sm group'.

Examination of histological sections of *B. pfeifferi* taken 8 days after exposure to only *C. sukari* miracidia revealed the presence of sporocysts that had undergone little or no growth or development and had host hemocytes around them (*Figure 3c*). By contrast, *S. mansoni* sporocysts were much larger at the same age and germinal development was underway (*Figure 3d*). The success of *C. sukari* sporocysts in *B. pfeifferi* increased significantly (p=<0.0192) if *S. mansoni* miracidia were also allowed to infect the snails, particularly so if the *S. mansoni* exposures followed the exposure of snails to *C. sukari*. Importantly, this was accompanied by a sharp

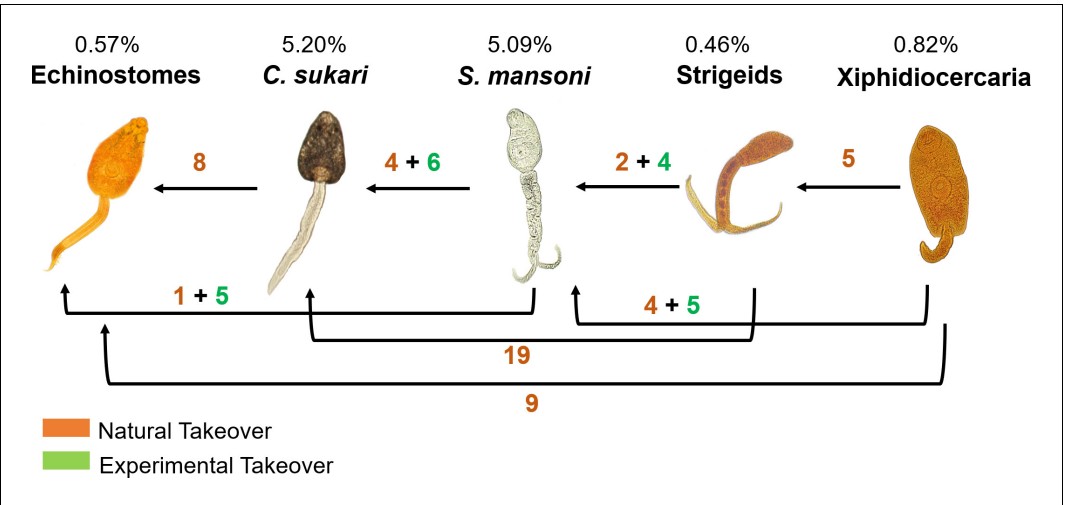

**Figure 2.** Trematode hierarchy in *B. pfeifferi* at Asao Stream. The dominance hierarchy was worked out through both experimental superinfections of snails with existing infections (green numbers), and by maintaining infected snails from the field to see if they switched from shedding one type of cercaria to another (orange numbers). Percentages shown are the prevalence of infected *B. pfeifferi* (shedding/total number of *B. pfeifferi* collected) for each group.

reduction in the number of snails shedding *S. mansoni* cercariae as compared to the snails exposed just to *S. mansoni* (*Figure 3b*).

A further indication of the antagonistic interactions between the two trematode species was provided by the observation that although infections of both *S. mansoni* and *C. sukari* were common in *B. pfeifferi* in west Kenyan stream habitats, double infections (snails simultaneously shedding cercariae of both species) were rare, and occurred less than expected by chance (p=<0.001) (*Figure 4a*). Some field snails found to be naturally shedding *S. mansoni* cercariae would, upon further observation in the lab, permanently switch over to producing *C. sukari* cercariae instead. Similarly, exposure of field-collected snails shedding *C. sukari* cercariae to *S. mansoni* miracidia rarely resulted in conversion of the infections to production of *S. mansoni* cercariae. However, exposure of snails that were shedding *S. mansoni* cercariae to miracidia of *C. sukari* more often resulted in production of *C. sukari* cercariae. Lastly, our survey data also showed that numbers of *S. mansoni* and *C. sukari* infections were positively correlated (r = 0.622) (*Figure 4b*). This is not surprising, since *C. sukari* relies on *S. mansoni* for its own transmission.

## Mathematical model

To estimate the potential impact of *C. sukari* on *S. mansoni* transmission, we developed a mechanistic mathematical model that explicitly accounts for parasite-parasite interactions within *B. pfeifferi* and is parameterized using values obtained from our experimental and field studies (supplementary data). In particular, our modeling framework tracks the outcomes for each parasite in the case of both simultaneous and sequential infections of *B. pfeifferi* with *C. sukari* and *S. mansoni*, based on survival rates from our empirical data. Furthermore, the model includes two snail size classes to account for size-dependent variation in snail survival and fecundity and cercariae output, which we have previously shown is important for accurately assessing the risk of infection to humans (Anderson et al., submitted). By varying the input of either *S. mansoni* or *C. sukari* miracidia to the model, we can examine how the relative abundance of *S. mansoni* to *C. sukari* larvae in the environment impacts the production of new parasites within the snail portion of the transmission cycle. The model predicts that the force of infection of *S. mansoni,* as measured by the proportion of cercariae-producing (shedding) snails, is reduced by approximately half in the presence of the antagonist *C. sukari* (*Figure 5a*). Only inputs of miracidia biased unrealistically heavily towards *S. mansoni* over *C. sukari* permit the highest numbers of *S. mansoni* cercariae to be produced (*Figure 5b*). We assume that

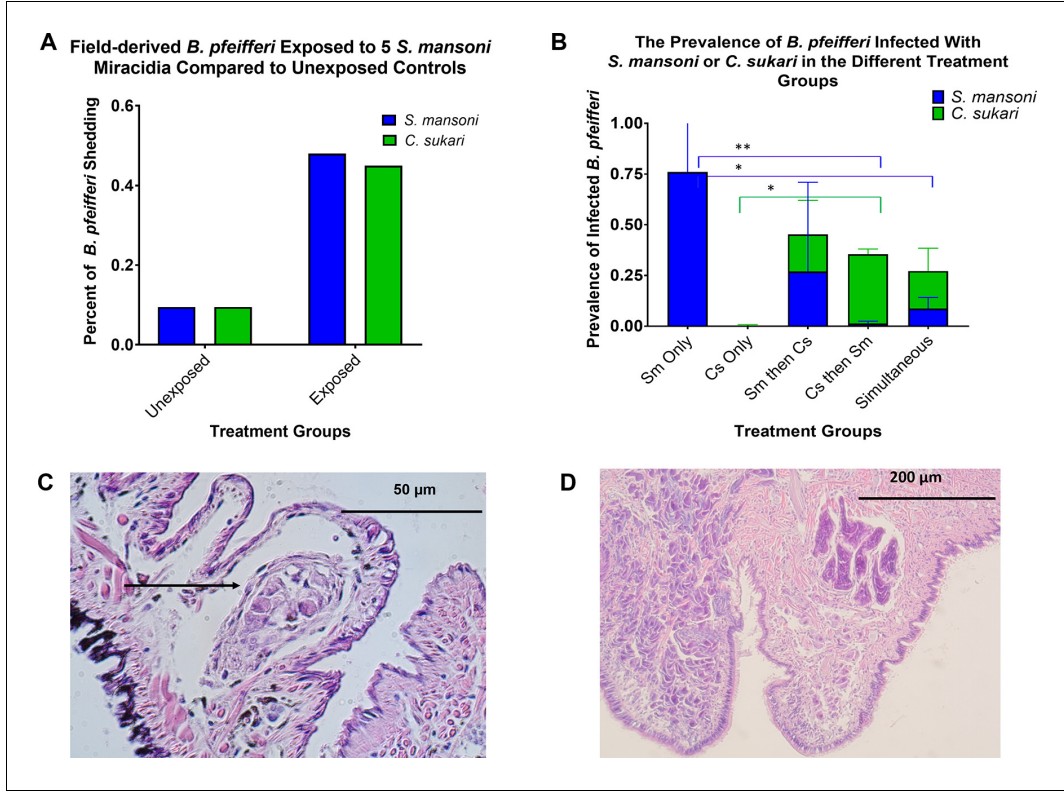

**Figure 3.** Human and cattle parasite interactions at Asao Stream. (**A**) Field-derived *B. pfeifferi* shown not to be shedding any cercariae at the time of collection were either left as unexposed controls or were exposed to *S. mansoni* (five miracidia/snail). Note that, unexpectedly, exposed snails were just as likely to subsequently shed *C. sukari* as *S. mansoni* cercariae compared to the control groups (p=<0.001). A few unexposed snails also shed cercariae, indicating that some of the snails had prepatent snails at the time of infection. (**B**) The prevalence of lab-reared *B. pfeifferi* exposed to various combinations of miracidia (see horizontal axis) of *C. sukari* and/or *S. mansoni* that subsequently shed cercariae of either species. Exposures to either species were with five miracidia/snail, 50 or 60 snails were used for each of 5 treatments for three separate experiments (total of 850 snails used). Separate ANOVAs were done for *S. mansoni* and *C. sukari* (each involving comparison of four groups), followed by pairwise comparisons. (**C**) Histological section of *B. pfeifferi* exposed to *C. sukari* for 8 days. Note the undeveloped sporocyst and the layer of hemocytes around it. (**D**) Histological section of *B. pfeifferi* exposed to *S. mansoni* for 8 days. The sporocyst has grown considerably and has developing daughter sporocysts within.

the presence of *C. sukari* infections in *B. pfeifferi* represent 'lost' *S. mansoni* infections. Therefore, our survey results (*Figure 2c*) suggest that if *C. sukari* were absent then the prevalence of *S. mansoni* shedders in streams would increase by 50% or more. Also, the model predictions highlight that, dependent on the background levels of infection, there may be increases of several hundred percent of *S. mansoni* cercariae in the absence of *C. sukari*.

## Discussion

*Schistosoma mansoni* is common in sub-Saharan Africa, aided by multiple transmission options, including humans, rodents and baboons as definitive hosts and by the widespread presence of several species of *Biomphalaria* vector snails occupying diverse aquatic habitats. Additionally, open human defecation and inadequate sanitation ensure widespread contamination of freshwater habitats with *S. mansoni* eggs (*Nagi et al., 2014*). We discovered at least 29 additional species of digenetic trematodes cycling through wild and domestic vertebrate definitive hosts that depend on *Biomphalaria* snails in western Kenya, creating inevitable opportunities for interactions if the larvae of two or more species co-occur in the same snail.

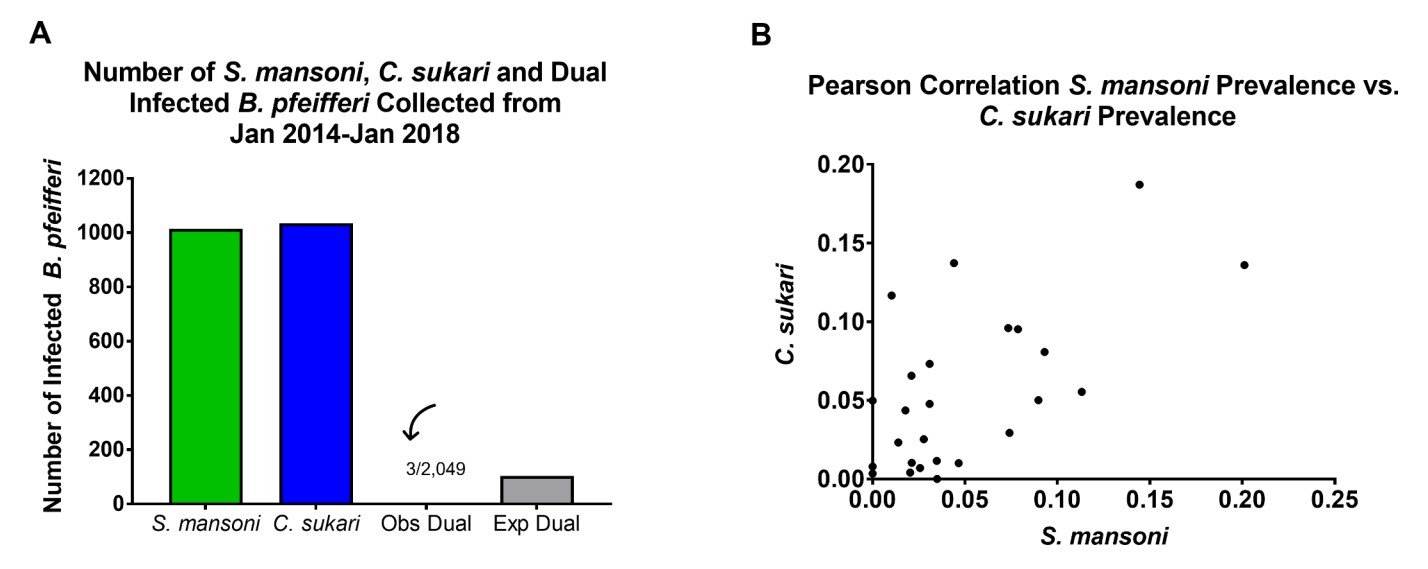

**Figure 4.** Field observations supported experimental results. (**A**) Graph showing the number of single infections of *B. pfeifferi* with either *S. mansoni* or *C. sukari* and the number of observed double infections, which was significantly fewer than the number of double infections expected by chance (p=<0.001). (**B**) Correlation between the abundance of *S. mansoni* and *C. sukari* infections (XY pairs = 25, Pearson r = 0.622, p=0.0009) in *B. pfeifferi* from our stream survey data (prevalence of each parasite from 25 collection time points).

In the rural settings where *S. mansoni* thrives, people often stand side-by-side with domestic ruminants to access water in streams or other water bodies, water that contains schistosome-infected *B. pfeifferi*. Domestic ruminants also predictably pass huge quantities of helminth eggs into the water, including those of *C. sukari* (we estimate ~10,000 eggs/cowpat). As *C. sukari* also develops in *B. pfeifferi*, a common and recurrent situation arises in which these two digenetic trematode species vie for access to their shared obligatory snail host.

We have shown that *S. mansoni* occupies an intermediate position in a trematode dominance hierarchy in *B. pfeifferi* in west Kenya and can be routinely displaced if present in co-infections by echinostomes and amphistomes, although particular species may vary in their predatory tendencies and the degree of their dominance (*Hechinger et al., 2011*; *Garcia-Vedrenne et al., 2016*). Surprisingly, even though much of the previous work focusing on interspecific trematode antagonism has been undertaken in the context of exploiting it as a potential control strategy for schistosomiasis (*Combes, 1982*; *Moravec et al., 1974*; *Pointier and Jourdane, 2000*; *Toledo and Fried, 2011*) ours is the first study undertaken in which all the species involved are extant in sub-Saharan Africa, where schistosomiasis assumes its greatest public health significance.

With respect to this hierarchy, the dominant trematodes are not the species we most commonly recovered from snails, for at least two reasons. One is that co-infections by no means inevitably occur within snails so that a species with a subordinate position is not always required to compete. This is of relevance for a species like *S. mansoni* occupying an intermediate hierarchical position in that it is not inevitably confronted in snails by predatory or inhibitory rediae of other species. Secondly, input of eggs from competitively dominant species like echinostomes often originates from birds (*Laidemitt et al., 2019*) and is likely dwarfed by the input of eggs from large definitive hosts like people, goats or cattle. This helps explain how the prevalence achieved by *S. mansoni* or by *C. sukari* is much higher than achieved by echinostomes despite the latter's competitive dominance in snails.

From the point of view of *S. mansoni*, colonizing a snail harboring a preexisting trematode infection, including even those snails infected with subordinate trematodes, is bound to interfere with its transmission to some extent. Successful infection will either be preempted by presence of a dominant species, or if take-over of a subordinate species occurs, the process likely takes weeks. In the

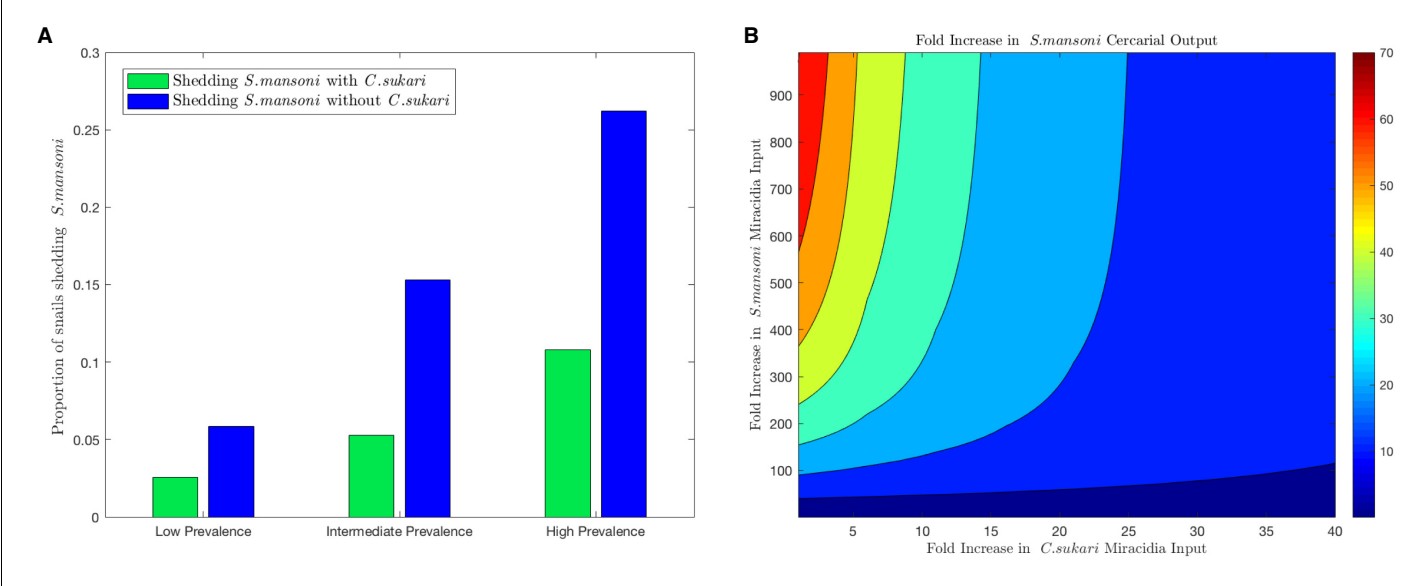

**Figure 5.** Mathematical model combining field and experimental observations. (**A**) At three different levels of *S. mansoni* prevalence in snails, the predicted reduction in prevalence of shedding snails from our transmission model due to the presence of *C. sukari* is estimated. In each case, the proportion of snails shedding *S. mansoni* is reduced by at least one half in the presence of *C. sukari*. (**B**) Relationship from model showing how *S. mansoni* cercariae production is maximized in this system only when the input of *S. mansoni* miracidia is very high relative to the input of *C. sukari* miracidia.

The online version of this article includes the following figure supplement(s) for figure 5:

**Figure supplement 1.** Conceptual diagram of the mathematical model.

**Figure supplement 2.** The proportion of snails shedding *S. mansoni* and *C. sukari* under a range of miracidial inputs and exposure rates.

**Figure supplement 3.** The infection prevalence of *S. mansoni* and *C. sukari* in *B. pfeifferi* and their cercarial production are sensitive to both the exposure rate per week per snail density in the habitat and the weekly input of *S. mansoni* miracidia (3a), but less sensitive to the weekly input of *C. sukari* miracidia (3b).

interim, production of *S. mansoni* cercariae will be delayed or reduced, and the infected snail could well expire before the take-over is complete.

Of the many trematodes coexisting with *S. mansoni*, *C. sukari* is most noteworthy for it is both abundant and dominant. Furthermore, the unusual facilitating effect provided to *C. sukari* by *S. mansoni* that we observed carries some provocative consequences. For example, from our experimental exposures of field-derived snails we found that *B. pfeifferi* were already colonized by *C. sukari* larvae, but the larvae were unable to complete development without a follow-up exposure to *S. mansoni*. This implies that most of the *C. sukari* infections we observed in streams represent snails that had been penetrated by *S. mansoni* miracidia and in which some degree of schistosome sporocyst development took place: the sporocysts were either stymied early in their development by the pre-existing presence of *C. sukari* larvae, or were able to develop to the point of cercariae production only to be taken over by a subsequent *C. sukari* infection. It is possible that other trematode species like strigeids or xiphidiocercariae may also facilitate *C. sukari* infections though they are much less abundant in our stream habitats than *S. mansoni*.

By using our empirical work to parameterize a novel mathematical model that explicitly accounts for interaction between parasites within snail hosts, we were able to quantify the role of antagonistic trematodes in field snails on the force of schistosomiasis transmission in an endemic setting. Specifically, our mathematical model predicted that *C. sukari* could dampen the force of transmission of *S. mansoni* by as much as 65 percent. The presence of *C. sukari*-infected cattle thereby provides a significant protective effect for humans. Furthermore, the amphistome would not persist or would persist at much lower levels without the facilitation provided by *S. mansoni*. Consequently, in current mass drug administration (MDA) campaigns in Kenya, treating humans with praziquantel (with

presumed downstream effects on limiting *S. mansoni* infections in snails) would then also decrease the number of successful *C. sukari* infections in snails and in turn the number of cattle infected with *C. sukari*. Conversely, if cattle were targeted in MDA campaigns to control *Fasciola gigantica* and other flukes like *C. sukari,* we would expect assuming no concomitant *S. mansoni* control, to observe an increase in the number of *S. mansoni* infections in snails. In particular, our model predicts at least a 40% increase (*Figure 5a*). The peculiar dependencies between *C. sukari* and *S. mansoni* provide an important example of how intramolluscan encounters could have significant downstream effects on the prevalence of infection of either parasite in their respective definitive hosts.

The interactions we have noted between *S. mansoni* and *C. sukari* favor the latter species, but in other contexts, including in Kenya, the presence of another amphistome species *Calicophoron microbothrium* in the snail *Bulinus tropicus* facilitates infections with the cattle schistosome *Schistosoma bovis* (*Southgate et al., 1989*). Schistosomes and amphistomes, possibly because their respective positions in the dominance hierarchy lie close together (*Figure 2*), clearly engage in distinctive interactions that seem to favor the colonization of a snail species that might otherwise not easily be infected.

Recent transcriptional studies have indicated that *S. mansoni* sporocysts promote down-regulation of many *B. pfeifferi* immune factors, and that *C. sukari* transcriptomic activity may be limited without *S. mansoni* present (*Buddenborg et al., 2017*). This is suggestive that immune interference provided by *S. mansoni* enables larval development of *C. sukari* to proceed (*Lim and Heyneman, 1972*). This idea is supported by finding, following experimental infections of *B. pfeifferi* by *C. sukari*, intact but undeveloped mother sporocysts of *C. sukari* loosely surrounded by snail hemocytes, as if parasite and host were at an impasse. Furthermore, several studies in the related snail, *Biomphalaria glabrata* indicate *S. mansoni* and *Echinostoma paraensei* can interfere with snail defenses, thereby allowing development of particular trematode strains and species that would otherwise be doomed to fail in *B. glabrata* (*Lie, 1982*; *Lie et al., 1976*; *Hanington et al., 2012*). We also note that immune-suppression mediated by one trematode species may be a means whereby another trematode species is able to invade and colonize an otherwise incompatible snail host, thereby helping to account for high rates of snail host-switching that are a prominent feature of trematode-snail interactions (*Brant and Loker, 2013*) and *C. sukari* may exemplify this process.

Also, of interest is the mechanism whereby subsequent development of *S. mansoni* sporocysts is suppressed in the presence of *C. sukari* larvae. Is it mediated directly by predatory or secretory activities of *C. sukari* larvae, or indirectly via snail host responses? Our histological and in vitro encounter studies do not reveal direct evidence of predatory activity by *C. sukari* rediae. It also seems unlikely that if *S. mansoni* initially provided immune-mediated protection for *C. sukari* that later in the co-infection the tables would be turned such that *S. mansoni* rather than *C. sukari* would be selectively attacked by host immune responses. One possibility to pursue is that secretory/excretory products produced by *C. sukari* larvae interfere with development of *S. mansoni* sporocysts, possibly including damaging them such that they are prone to attack by hemocytes of *B. pfeifferi*. Identification of factors inimical to *S. mansoni* larval development would be of potential interest as a novel means of schistosomiasis control. We further note that in an area of hyperendemic transmission of *S. mansoni* in and around Lake Victoria, this predominantly human parasite is nonetheless favored by its ability to infect diverse definitive hosts and at least three taxa of ecologically distinct *Biomphalaria* intermediate hosts. However, *S. mansoni* transmission is modulated by a number of biological realities that relate both to historical patterns in animal husbandry favoring transmission of trematodes of domestic ruminants and to the presence of several wild vertebrate species that support a remarkably diverse trematode fauna. Many of these trematodes, at least 29 species in addition to *S. mansoni*, also depend on *Biomphalaria* snails for their larval development. Our results indicate that *S. mansoni* is in no way favored by their presence but must interact with them if found in the same snail host and can even be directly exploited by them by providing a needed facilitation effect, one that favors the continued presence and abundance of a potent competitor, the most common trematode in the system, the cattle-transmitted *C. sukari*.

In conclusion, the peculiar interactions we found between *S. mansoni* and *C. sukari* highlight the important role of human-modified history in dictating patterns of infectious disease. The long-term use of the same snail-containing aquatic habitats by human pastoralists and their livestock has led to a situation where the parasites of cattle have seemingly been selected to exploit the predictable and widespread presence of the human parasite *S. mansoni* in snails. Here we show another example of

how pastoralism has shaped the nature of the parasites and diseases experienced by both humans and ruminants (*Greter et al., 2017*). In addition, the interspecific interactions occurring between trematodes within their vector snails can in turn have cascading effects in influencing prevalence of parasites in their definitive hosts, including humans.

## Materials and methods

### Field surveys and parasite assessment

To measure transmission patterns of trematode parasites in their snail hosts, we sampled six sites along a 300 m stretch of Asao Stream (−0.31810, 35.00690) in western Kenya every other month from January 2014-January 2018. We also sampled six sites along a 300 m stretch at Kasabong Stream (−0.15190, 34.33550) and two sites along the shoreline at Lake Victoria (−0.09410, 34.70760) from Jan 2015-Jan 2018. At the stream and shoreline sites, two people sampled for 15 min using a long-handled triangular mesh scoop, scooping the sides of rocks, plants, and the bottom of the substrate to collect snails. Water velocity and pH (Hanna Instruments pH/Conductivity/TDS High-Range Tester) measurements were also recorded. Air temperature and rainfall data were collected from the Kisumu, Kenya airport weather station. Snails were then transported back to the lab and following cleaning and sorting were individually placed into 12-well tissue culture plates with 3 ml of aged tap water. The plates were then placed in natural light for 2 hr to induce the release of cercariae ('shedding'). Snails were identified using an African snail key (*Brown and Kristensen, 1989*) and cercariae were provisionally identified using African and North American keys (*Frandsen and Christensen, 1984*; *Schell, 1985*). Some snails found to be infected with trematodes were used for experimental exposures to determine which trematodes were dominant or subordinate to *S. mansoni* (see below). Uninfected snails were placed into 20 L plastic aquaria and fed red leaf lettuce and then re-shed 3 weeks post-collection to determine if they had harbored prepatent trematode infections (incompletely developed infections) at the time of collection. Most cercariae from each individual shedding snail were saved in a separate 2 ml screwcap tube and preserved in 95% ethanol for later molecular analyses.

Because cryptic species commonly exist among trematodes, we employed molecular markers to more precisely differentiate species in conjunction with morphological features of the cercariae. We used both nuclear markers 28S ribosomal gene (28S) or intragenic spacer (ITS), and mitochondrial markers cytochrome oxidase 1 (*cox*1) or NADH dehydrogenase 1 (*nad*1) for this purpose. The choice of molecular markers used depended upon the number of GenBank records already available for each trematode superfamily. For example, there was a greater diversity of GenBank records for ITS than for the 28S gene for amphistomes. Genomic DNA was extracted from 1 to 3 cercariae using the Qiagen MicroKit (Qiagen, Valencia CA), with a final elution of 35 µl. PCR cycles and primers used were as reported in *Laidemitt et al. (2017)* for ITS and *cox*1 for amphistomes, *Morgan and Blair (1998)* for the *nad*1 gene for echinostomes, *Tkach et al. (2016)* for the 28S gene which amplifies many superfamilies of trematodes and (*Lockyer et al., 2003*) for *cox*1 to amplify schistosomes. PCR products were purified using ExoSap-IT (Affymetrix, Santa Clara, CA).

PCR amplicons were sequenced in both directions using an Applied Biosystems 3100 automated sequencer and BigDye terminator cycle sequencing kit Version 3.1 (Applied Biosystems, Foster City, CA). Sequences were edited in Sequencher 5.0 and then were aligned by eye in Mega 7 (*Kumar et al., 2016*) against the diversity of sequences in GenBank for each gene. More details about these methods are described in *Laidemitt et al. (2017)* and *Laidemitt et al. (2019)*. To determine species or genus to which our samples were most closely aligned, we ran pairwise comparisons in Mega 7. Operationally, we used *p-distance* values less than 1.5% with respect to *nad*1 or *cox*1 genes to indicate our samples corresponded to those of a single species, and values > 5% to indicate they were different species (*Vilas et al., 2005*).

### Compatibility of *Biomphalaria* to *S. mansoni*

Fifty F1 *Biomphalaria pfeifferi* or *B. choanomphala*, or 60 *B. sudanica* were individually exposed to 5 *S. mansoni* miracidia from sympatric localities (Asao for *B. pfeifferi*, Kanyibok for *B. sudanica* and *B. choanomphala*) in 12 well cell culture plates for 6 hr and then placed into 20 L tanks and fed lettuce and shrimp pellets three times a week. Water was changed once a week. Snails were

put in 12 well plates under ambient light to induce shedding once a week starting 3 weeks post exposure. Peak shedding prevalence of surviving snails, which is the number shedding *Biomphalaria*/surviving *Biomphalaria* are shown (*Figure 1D*). *B. pfeiffieri* shed the earliest (4 weeks and peaked at 6 weeks) and *B. sudanica* and *B. choanomphala* started shedding at 6 weeks and peaked at 7 weeks. Snails used in this experiment were identified by reference to the mitochondrial genomes of *Biomphalaria* collected from the same locality (Asao) or nearby localities (Kanyibok) and published by *Zhang et al. (2018)*.

## Experimental exposures to examine interactions between *S. mansoni* and the amphistome *Calicophoron sukari*

Experiments were set up to examine the interactions between *S. mansoni* and the common amphistome we provisionally identified as *C. sukari* (*Dinnik, 1954*). *S. mansoni* eggs were collected and pooled from five primary school children (see ethics statement below) and *C. sukari* eggs were harvested from cow dung samples collected along the banks of Asao stream. Eggs from both sources were concentrated using nested sieves (*Mutuku et al., 2014*). After the cattle dung was sieved, half of the retained egg-rich material was placed into the refrigerator as a source of *C. sukari* eggs for later parts of the same experiment, and the other half was placed in plastic containers and aerated in the dark for 14–16 days to allow the eggs to fully embryonate. Eggs were considered fully embryonated when moving miracidia were observed within the eggs.

To establish F1 *B. pfeifferi* for the experiment, at least 200 uninfected (screened by shedding) parental snails were collected from Asao stream or Kasabong stream and divided among five 180 L tanks for breeding. Snails were fed red leaf lettuce and shrimp pellets three times per week until eggs were laid.

Fifty (2014 experimental exposures) or sixty (2017 experimental exposures) 2–4 mm laboratory-reared F1 *B. pfeifferi* (parents from Kasabong stream in 2014 and parents from Asao stream in 2017) were assigned to each of the following six treatment groups: 1) sham controls (no parasite exposure, but individually placed into 12 well plates); 2) snails individually exposed to 5 *S. mansoni* miracidia; 3) snails individually exposed to 5 *C. sukari* miracidia; 4) snails individually exposed to 5 *S. mansoni* miracidia and two weeks later to 5 *C. sukari* miracidia; 5) snails individually exposed to 5 miracda of *C. sukari* then two weeks later to five miracidia each of *S. mansoni*; and 6) snails simultaneously exposed to 5 *C. sukari* and 5 *S. mansoni* miracidia. F1 snails were randomly chosen from the five 180 L tanks and placed individually into 12-well cell culture plates with 3 ml of aged tap water for the exposures. Miracidia were hatched by placing a flask or plastic container with *S. mansoni* or *C. sukari* eggs in water in ambient light. Miracidia were transferred using a glass pipette into a petri dish to facilitate enumeration and were then pipetted into the 12-well cell culture plates (five miracidia per species per well). All snails were exposed to miracidia obtained within 2 hr of hatching and were exposed for 6 hr in the 12 well plates. Snails were then placed into 20 L tanks in a screened, covered snail rearing facility subject to the diurnal light and temperature at Kisian, Kenya. The snails were kept in 20 L tanks and were fed red leaf lettuce and shrimp pellets three times a week. Water was changed once a week.

Starting at three weeks post-exposure and at weekly intervals thereafter, snails were individually placed into wells in 12-well cell culture plates with 3 ml of aged tap water for two hours between 10:00-12:00 under ambient light to determine if they would shed cercariae. Cercariae were identified according to cercariae keys (*Schell, 1985*) and for verification and further delineation, some cercariae were sequenced (GenBank accession numbers MN603500-MN603506). Snails were shed once a week until 10 weeks post second parasite exposure (12 weeks from initial exposure). The number of shedders and snail mortality were recorded and prevalence of infection determined as described above. This basic experimental protocol was repeated three times, in Jan 2014, Jan 2017 and Jun 2017.

An appropriate sample size was computed when the study was being designed. We used G*Power to calculate sample size with an alpha error probability of 0.05 and found our sample size was appropriate (250 to 300 for each experiment or 50 to 60 per each treatment). Differences among treatment groups with respect to infection success of experimental exposures were analyzed GraphPad Prism 7 for both *S. mansoni* and *C. sukari* using a non-parametric Kruskal-Wallis analysis of variance (ANOVA) with an alpha of 0.05, by exposure type (single parasite, and mixed species co-exposure), followed by pair-wise comparisons. In the co-exposure

experiments we used shedding rates at 8 weeks post second parasite exposure (10 weeks post exposure for controls). Non-parametric tests were used because the data were not normally distributed because survivorship and infection prevalence varied among the infection groups. Prevalence of infection of each parasite per group was determined as the proportion of surviving exposed *B. pfeifferi* shedding cercariae.

## Assessing relative dominance ranking among trematodes

Two different methods were used to reveal the trematode dominance hierarchy. The first used field-derived *B. pfeifferi* found to be shedding one species of cercaria. These were kept in aquaria and were shed twice a week until the snail died. A 'natural takeover' was recorded if the snail ceased shedding one species of cercaria and switched over to shedding a different species of cercaria (for example, *B. pfeifferi* was first shedding *S. mansoni* and later began shedding *C. sukari* cercariae). The second method was to obtain either field-derived *B. pfeifferi* shedding one type of cercaria, or to experimentally expose lab-reared *B. pfeifferi* to a particular species of trematode. Then, once the snails were shedding cercariae of that species, they were re-exposed to miracidia of a different species of trematode. These snails were isolated and shed twice a week starting three weeks post second exposure to learn if cercariae of the second species were produced.

## Histology

Snails were placed in Railliet-Henry's fluid for at least 48 hr. The shell of each snail was removed, and the body of the snail was placed into 10% neutral buffered formalin. The snails were sent to TriCore Reference Laboratories in Albuquerque, New Mexico, sectioned, and sections stained with hematoxylin and eosin.

## Mathematical modeling methods

We developed a deterministic model framework, described by a system of ordinary differential equations, to evaluate the impact of *C. sukari* presence and transmission intensity on *S. mansoni* prevalence in *B. pfeifferi* and total *S. mansoni* cercarial production (*Figure 5—figure supplement 1* and *Supplementary file 3*). The model divides the *B. pfeifferi* snail population into two size classes (juvenile and adults) and multiple infection stages (susceptible, exposed, infected, infected and castrated) to account for the reduction in growth rates and fecundity and an increase in mortality associated with *S. mansoni* and *C. sukari* infections. It also incorporates the timing of each trematode infection (both simultaneous and sequential) on consequent *S. mansoni* or *C. sukari* cercarial output as determined by our laboratory experiments. In addition, the model includes a 'sink' snail population to account for the loss of miracidia (of both trematode species) when the larvae infect incompetent snail species that do not result in new cercariae. Moreover, the model explicitly tracks the loss of miracidia due to the successful infection of any snail host.

The equilibrium levels of *S. mansoni* and *C. sukari* infection were calibrated to field survey data by fixing the rate of snail exposure per miracidial density and the *S. mansoni* and *C. sukari* miracidia input into the system. Field survey data, obtained from January 2014-January 2018 at Asao stream in western Kenya, identify the proportion of *B. pfeifferi* shedding each trematode. These data indicate that the proportion of *B. pfeifferi* shedding *S. mansoni* was equivalent to the proportion shedding *C. sukari*. We set miracidial inputs and exposure rates to yield three different *S. mansoni* and *C. sukari* snail shedding regimes, at 2.5, 5 and 10 percent. The relative size of the exposure rates reflects miracidial preference for *B. pfeifferi* over potential sink snails from our choice chamber experiments, and the unlikely possibility of simultaneous exposure. The volume of water in the system and the overall snail density and relative abundance of *B.pfeifferi* and potential sink snails were obtained from snail density surveys conducted from June 2016 to January 2018 on a 380 meter section of Asao stream. The impact of varying levels of *C. sukari* miracidial input into the *S. mansoni* transmission system at equilibrium was then evaluated with respect to the prevalence of *S. mansoni* infection in *B. pfeifferi* and the overall *S. mansoni* cercarial output of the snail population. The sensitivity of *B. pfeifferi* infection with *S. mansoni* and *C. sukari* and resulting cercarial output to miracidial input and exposure rate of *B. pfeifferi* per miracidial density was explored and is represented in *Figure 5—figure supplements 2* and *3*. At the snail density we found at Asao stream the miracidial input values that resulted in equal shedding of *S. mansoni* and *C. sukari* at reasonable inputs, of

between 50–5,000,000 miracidia per week, were only possible in certain exposure ranges. These *B. pfeifferi* exposure rates per miracidial density per week were primarily between 0.025 and 0.75 (*Figure 5—figure supplement 1*). All model variables, differential equations and parameter values are also described in *Supplementary files 1–3*. All model simulations were performed in Matlab 2019a (The MathWorks Inc, Natick, MA, 2019).

## Ethical approval

Human subjects were enrolled into our study to provide fecal samples containing *Schistosoma mansoni* eggs. We used these eggs to do our experimental exposures and to develop the dominance hierarchy. Samples were collected and pooled from five primary school children from Obuon primary school near Asao Stream, Kenya (00°19'01' S, 035°00'22' E). Consent forms were given and signed by the children's parents. The KEMRI Ethics Review Committee (SSC No. 2373) and the UNM Institution Review Board (IRB 821021–1) approved all aspects of this project involving human subjects. All children tested and found positive for *S. mansoni* were treated with praziquantel following standard protocols. Details of recruitment and participation of human subjects for fecal collection are described in *Mutuku et al. (2014)*. This project was undertaken with the approval of Kenya's National Commission for Science, Technology, and Innovation (permit number NACOSTI/P/15/9609/4270), National Environment Management Authority (NEMA/AGR/46/2014) and cercariae and snails were exported with the approval of the Kenya Wildlife Service permit number 0004754.

## Acknowledgements

We thank Dr. Sarah Buddenborg, Dr. Eric Lelo, Joseph Kinuthia, Geoffrey Maina, Boaz Oduor and Ibrahim Mwangi for assistance with collection of field samples and snail maintenance and Dr. Jennifer Rudgers for her statistical expertise. We also thank the International Livestock Research Institute (ILRI), Nairobi, Kenya for sequencing some of our samples. Technical assistance at the University of New Mexico Molecular Biology Facility was supported by the National Institute of General Medical Sciences of the National Institutes of Health under Award Number P30GM110907. We gratefully acknowledge the following agencies for their financial support: The National Institute of Health (NIH) grant R37AI101438, the Fogarty International Center and National Institute of Mental Health, NIH award number D43TW010543, and the Bill and Melinda Gates Foundation, Seattle, WA (OPP1098449) for the Grand Challenges Explorations Initiative grant. The content for this paper is solely the responsibility of the authors and does not necessarily represent the official views of the National Institutes of Health. This paper was published with the approval of the Director, of KEMRI.

## Additional information

### Funding

| Funder | Grant reference number | Author |
|---|---|---|
| National Institutes of Health | R37AI101438 | Gerald M Mkoji<br>Eric S Loker |
| Fogarty International Center | D43TW010543 | Martina R Laidemitt |
| Bill and Melinda Gates Foundation | OPP1098449 | Eric S Loker |

The content for this paper is solely the responsibility of the authors and does not necessarily represent the official views of the National Institutes of Health.

### Author contributions

Martina R Laidemitt, Conceptualization, Data curation, Formal analysis, Validation, Investigation, Visualization, Methodology, Writing—original draft, Writing—review and editing; Larissa C Anderson, Data curation, Software, Formal analysis, Methodology, Writing—original draft, Writing—review and editing; Helen J Wearing, Formal analysis, Validation, Methodology, Writing—original draft, Writing—review and editing; Martin W Mutuku, Data curation, Investigation, Methodology,

Writing—original draft, Writing—review and editing; Gerald M Mkoji, Resources, Supervision, Funding acquisition, Writing—original draft, Project administration, Writing—review and editing; Eric S Loker, Conceptualization, Resources, Data curation, Supervision, Funding acquisition, Investigation, Methodology, Writing—original draft, Writing—review and editing

### Author ORCIDs
Martina R Laidemitt (ID) https://orcid.org/0000-0001-6250-5555
Larissa C Anderson (ID) https://orcid.org/0000-0002-4259-4068
Helen J Wearing (ID) https://orcid.org/0000-0002-9837-9797

### Ethics
Human subjects: Human subjects were enrolled into our study to provide fecal samples containing Schistosoma mansoni eggs. We used these eggs to do our experimental exposures and to develop the dominance hierarchy. Samples were collected and pooled from five primary school children from Obuon primary school near Asao Stream, Kenya (00°19'01" S, 035°00'22" E). Consent forms were given and signed by the children's parents. The KEMRI Ethics Review Committee (SSC No. 2373) and the UNM Institution Review Board (IRB 821021-1) approved all aspects of this project involving human subjects. All children tested and found positive for S. mansoni were treated with praziquantel following standard protocols. Details of recruitment and participation of human subjects for fecal collection are described in Mutuku et al. 2014.

### Decision letter and Author response
Decision letter https://doi.org/10.7554/eLife.50095.sa1
Author response https://doi.org/10.7554/eLife.50095.sa2

## Additional files
### Supplementary files
- Supplementary file 1. Description of model variables.
- Supplementary file 2. Description of parameter values.
- Supplementary file 3. Model equations.
- Transparent reporting form

### Data availability
GenBank records: MN603500-MN603506.

The following datasets were generated:

| Author(s) | Year | Dataset title | Dataset URL | Database and Identifier |
|---|---|---|---|---|
| Martina R Laidemitt, Larissa C Anderson, Helen J Wearing, Martin W Mutuku, Gerald M Mkoji, Eric S Loker | 2019 | Calicophoron sukari isolate PA121 cytochrome c oxidase subunit I (cox1) gene, partial cds; mitochondrial | https://www.ncbi.nlm.nih.gov/nuccore/MN603500 | NCBI GenBank, MN603500 |
| Martina R Laidemitt, Larissa C Anderson, Helen J Wearing, Martin W Mutuku, Gerald M Mkoji, Eric S Loker | 2019 | Calicophoron sukari isolate PA122 cytochrome c oxidase subunit I (cox1) gene, partial cds; mitochondrial | https://www.ncbi.nlm.nih.gov/nuccore/MN603501 | NCBI GenBank, MN603501 |
| Martina R Laidemitt, Larissa C Anderson, Helen J Wearing, Martin W Mutuku, Gerald M Mkoji, Eric S Loker | 2019 | Calicophoron sukari isolate PA123 cytochrome c oxidase subunit I (cox1) gene, partial cds; mitochondrial | https://www.ncbi.nlm.nih.gov/nuccore/MN603502 | NCBI GenBank, MN603502 |

| | | | | |
|---|---|---|---|---|
| Martina R Laidemitt, Larissa C Anderson, Helen J Wearing, Martin W Mutuku, Gerald M Mkoji, Eric S Loker | 2019 | Calicophoron sukari isolate PA124 cytochrome c oxidase subunit I (cox1) gene, partial cds; mitochondrial | https://www.ncbi.nlm.nih.gov/nuccore/MN603503 | NCBI GenBank, MN603503 |
| Martina R Laidemitt, Larissa C Anderson, Helen J Wearing, Martin W Mutuku, Gerald M Mkoji, Eric S Loker | 2019 | Schistosoma mansoni isolate Sm1 cytochrome c oxidase subunit I (cox1) gene, partial cds; mitochondrial | https://www.ncbi.nlm.nih.gov/nuccore/MN603504 | NCBI GenBank , MN603504 |
| Martina R Laidemitt, Larissa C Anderson, Helen J Wearing, Martin W Mutuku, Gerald M Mkoji, Eric S Loker | 2019 | Schistosoma mansoni isolate Sm2 cytochrome c oxidase subunit I (cox1) gene, partial cds; mitochondrial | https://www.ncbi.nlm.nih.gov/nuccore/MN603505 | NCBI GenBank, MN603505 |
| Martina R Laidemitt, Larissa C Anderson, Helen J Wearing, Martin W Mutuku, Gerald M Mkoji, Eric S Loker | 2019 | Schistosoma mansoni isolate Sm3 cytochrome c oxidase subunit I (cox1) gene, partial cds; mitochondrial | https://www.ncbi.nlm.nih.gov/nuccore/MN603506 | NCBI GenBank, MN603506 |

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
