## [Decision Letter]

**Acceptance summary:**

Laidemitt et al. elegantly combine observational, experimental and modeling approaches to offer new insights into how parasite interactions can enhance or decrease transmission depending on the context. They studied human and cattle flatworms, which compete for the resources in a shared snail intermediate host when humans and their domestic animals use the same aquatic habitats. The human flatworm cause the debilitating neglected tropical diseases schistosomiasis. The authors discovered that efforts aimed at reducing the transmission of human schistosomes would help to also control the cattle parasite, whereas efforts targeted at the cattle parasite would be detrimental to schistosomiasis control, thus having important implications for the management of this widespread human disease.

**Decision letter after peer review:**

Thank you for submitting your article "Antagonism between parasites within snail hosts impacts the transmission of human schistosomiasis" for consideration by *eLife*. Your article has been reviewed by three peer reviewers, and the evaluation has been overseen by a Reviewing Editor and Neil Ferguson as the Senior Editor. The following individuals involved in review of your submission have agreed to reveal their identity: Bonnie Webster (Reviewer #1); Giulio De Leo (Reviewer #3).

The reviewers have discussed the reviews with one another and the Reviewing Editor has drafted this decision to help you prepare a revised submission.

Summary:

The reviewers and I found this to be an interesting and well-designed study. I particularly found the question important and understudied, and the findings potentially significant to schistosomiasis control. However, there are some issues that should be addressed that I do think will improve this manuscript. Please see below.

Essential revisions:

I received two reviews from reviewers with considerable experience in mathematical modeling of schistosomaisis. One suggested that some sensitivity and uncertainty analyses would improve the paper, and I agree. The second noticed a potential error in the model and both reviewers had concerns about model complexity. These should be addressed. Reviewer #1 raised concerns about clarity of the methods and the results. These concerns should also be addressed.

Reviewer #1:

This is a very interesting, important and well-executed project. It is also scientifically novel and presents some important results.

My major concern is some of the clarity of the methods and also the results read more like a discussion.

For the results, I suggest adding actually numbers rather than just an overview of the findings.

There are several important details missing from the results which are needed for clarity of the data and to see how the experiments were performed.

I have detailed these areas that need changing or adding too below:

Subsection “Field surveys and parasite assessment”: Also, how were the cercariae preserved i.e. individually, pooled and in how much ethanol. Also, relate this to the extraction. Individual or pooled cercariae extracted.

Subsection “*C. sukari* and *S. mansoni* Experimental Exposures”, paragraph two: to help the readers it would be good to have the primer and PCR details in the Materials and methods even though they are published.

Results section: generally, it would be good to have more numbers and actual amounts stated. I take the example of paragraph two in subsection “*C. sukari* and *S. mansoni* Experimental Exposures”: "Some field snails…", "rarely resulted…" it would be better to have an actual number of snails. At the moment the results read more like a discussion.

Subsection “Field surveys and parasite assessment” paragraph two: add how the sequences were edited before analysis.

The final two sentences of the same paragraph: this is confusing. Can you clarify? Did you use the pairwise comparisons to determine your species or sub-species clusters or determine the actual species? Did you have reference sequences in the analysis?

Subsection “Compatibility of *Biomphalaria* to *S. mansoni*”: in the results you do not refer to F1 snails in the experiments.

In the same section explain what you did at each exposure time. Did you record the numbers and then take those infected snails out of the experiment to get a final figure? How many weeks did you continue? Did you record deaths and how did you take these into account in your percentage infection calculations? 50 or 60 is too vague, how many were used? What is peak shedding prevalence? Also the final sentence of this subsection is actually results not methods so the structure needs changing. Also, here how were the different snails identified and how robust is that identification?

In subsection “Experimental exposures to examine interactions between *S. mansoni* and the amphistome *Calicophoron sukari*”: here you describe the rearing of the snails. But these lab. reared snails are used in the section above. Some restructuring is needed to allow the Materials and methods to be followed better.

Final sentence of the first paragraph in subsection “Experimental exposures to examine interactions between *S. mansoni* and the amphistome *Calicophoron sukari*”: is there a reference for this method?

Paragraph two: again 50-60 is too vague, add the actual number?

In the same paragraph is “in Jan2014” correct? that is a long time before the 2017 studies?

How many weeks after did you stop and were deaths recorded?

How were the cercariae identified?

There are no details on how the snails were individually exposed, in what kind of vessel, the general amount of water, for how long, how were the miracidia captured?

In the first paragraph of the same section: how do you know they are embryonated and ready for collection for the exposure experiment.

Subsection “Assessing relative dominance ranking among trematodes”: can you add some numbers of snails that were involved. Again for subsection “Histology” and which exposed combinations.

Reviewer #2:

This is a very interesting study, combining field observations, laboratory experiments, and mathematical models, to explore mechanisms of antagonism between different species of trematodes within snail hosts and the likely impacts on the transmission of human schistosomiasis. Overall, the study was well-designed and implemented, and novel, generating new insights into complexity of *S. mansoni* transmission and having implications on schistosomiasis control programs.

The only major comment relates to the mathematical modeling on which an important result (removal of *C. sukari* would increase *S. manosoni*-infected snail by three-fold) was based. The mathematical model has too many parameters, given the scale (e.g. limited numbers) of both field observations and lab experiments and the degree to which the majority of parameters relied on parameter estimations, it worried me a bit about the "uncertainty" of the result. It might be helpful to perform sensitivity and uncertainty analyses on the model. Other than that, this is a well-done study.

---

## [Author Response]

Essential revisions:[…]Reviewer #1:[…]Subsection “Field surveys and parasite assessment”: Also, how were the cercariae preserved i.e. individually, pooled and in how much ethanol. Also, relate this to the extraction. Individual or pooled cercariae extracted.

Details added: “Most cercariae from each individual shedding snail were saved in a separate 2 ml screwcap tube and preserved in 95% ethanol for later molecular analyses”. Next paragraph this was added: “Genomic DNA was extracted from 1-3 cercariae using the Qiagen MicroKit (Qiagen, Valencia CA).”.

Subsection “C. sukari and S. mansoni Experimental Exposures”, paragraph two: to help the readers it would be good to have the primer and PCR details in the Materials and methods even though they are published.

This would add significant text to the paper and the details are described in the papers that are cited within the text. If you find this to be necessary, we can add them in.

Results section: generally, it would be good to have more numbers and actual amounts stated. I take the example of paragraph two in subsection “C. sukari and S. mansoni Experimental Exposures”: "Some field snails…", "rarely resulted…" it would be better to have an actual number of snails. At the moment the results read more like a discussion.

Numbers of snails and more specific numbers have been added to the Results section as requested.

Subsection “Field surveys and parasite assessment” paragraph two: add how the sequences were edited before analysis.

Sequences were edited by eye in Sequencher 5.0 and then were aligned by eye in Mega 7 (Kumar et al., 2016) against the diversity of sequences in GenBank for each gene.

The final two sentences of the same paragraph: this is confusing. Can you clarify? Did you use the pairwise comparisons to determine your species or sub-species clusters or determine the actual species? Did you have reference sequences in the analysis?

To determine species or genus to which our samples were most closely aligned, we ran pairwise comparisons in Mega 7 against sequences for species that are in GenBank. Operationally, we used p-distance values less than 1.5% with respect to nad1 or cox1 genes to indicate our samples corresponded to those of a single species, and values >5% to indicate they were different species (Vilas et al., 2005).

Subsection “Compatibility of Biomphalaria to S. mansoni”: in the results you do not refer to F1 snails in the experiments.

Added F1 to the Results and Materials and methods.

“Fifty (2014 experimental exposures) or sixty (2017 experimental exposures) 2-4 mm laboratory-reared F1 *B. pfeifferi* (parents from Kasabong stream in 2014 and parents from Asao stream in 2017) were assigned to each of the following six treatment groups:”

In the same section explain what you did at each exposure time. Did you record the numbers and then take those infected snails out of the experiment to get a final figure?

All these data are now included in the Results and it is now clearer.

How many weeks did you continue? Did you record deaths and how did you take these into account in your percentage infection calculations? 50 or 60 is too vague, how many were used? What is peak shedding prevalence?

Data added.

Also the final sentence of this subsection is actually results not methods so the structure needs changing. Also, here how were the different snails identified and how robust is that identification?

We have taken methods from the Results section. Snails used in this experiment were in reference to the mitochondrial genomes of *Biomphalaria* collected from the same locality (Asao) or nearby localities (Kanyibok) and published by Zhang et al., 2016.

In subsection “Experimental exposures to examine interactions between S. mansoni and the amphistome Calicophoron sukari”: here you describe the rearing of the snails. But these lab. reared snails are used in the section above. Some restructuring is needed to allow the Materials and methods to be followed better.Final sentence of the first paragraph in subsection “Experimental exposures to examine interactions between S. mansoni and the amphistome Calicophoron sukari”: is there a reference for this method?

There is no reference for this method. This is something we had to figure out on our own.

Paragraph two: again 50-60 is too vague, add the actual number?

Actual numbers added.

In the same paragraph is “in Jan 2014” correct? that is a long time before the 2017 studies?

Yes, we repeated the experiment three times over different years. If anything, we feel that given the fact we could repeat this experiment and get the same outcome years later is a solidifying factor indicating this is not an uncommon event in these streams.

How many weeks after did you stop and were deaths recorded?How were the cercariae identified?

Information added in the Materials and methods section as requested.

There are no details on how the snails were individually exposed, in what kind of vessel, the general amount of water, for how long, how were the miracidia captured?

Information added as requested:

Miracidia were transferred using a glass pipette into a petri dish to facilitate enumeration and were then pipetted individually into the 12-well cell culture plates (5 miracidia per species per well). All snails were exposed to miracidia obtained within 2 hours of hatching and were exposed for 6 hours in the 12 well plates.

In the first paragraph of the same section: how do you know they are embryonated and ready for collection for the exposure experiment?

Eggs were considered fully embryonated when moving miracidia were observed within the eggs.

Subsection “Assessing relative dominance ranking among trematodes”: can you add some numbers of snails that were involved. Again for subsection “Histology” and which exposed combinations.

Information added as requested.

Reviewer #2:[...]The only major comment relates to the mathematical modeling on which an important result (removal of C. sukari would increase S. manosoni-infected snail by three-fold) was based. The mathematical model has too many parameters, given the scale (e.g. limited numbers) of both field observations and lab experiments and the degree to which the majority of parameters relied on parameter estimations, it worried me a bit about the "uncertainty" of the result. It might be helpful to perform sensitivity and uncertainty analyses on the model. Other than that, this is a well-done study.

We appreciate the reviewer’s concerns and have updated the sensitivity analyses. Figure 5—figure supplement 3 demonstrates how the major parameters that we estimated to calibrate the model against field survey data, the miracidial input of *C. sukari* and the exposure rate of *Biomphalaria* snails to *S. mansoni,* impact cercarial output and infection prevalence for both parasites. Observed prevalence levels (<10%) constrain the possible parameter combinations: if exposure rates are too low or too high, miracidial input has to be set to unrealistic levels. Figure 4 illustrates that the *Biomphalaria* exposure rate to *S. mansoni* and *S. mansoni* miracidial input have larger impacts on *S. mansoni* and *C. sukari* prevalence and shedding than *C. sukari* miracidial input.